# ADDRESSING REPRESENTATION COLLAPSE IN VECTOR QUANTIZED MODELS WITH ONE LINEAR LAYER

## ABSTRACT

Vector Quantization (VQ) is a widely used method for converting continuous representations into discrete codes, which has become fundamental in unsupervised representation learning and latent generative models. However, VQ models are often hindered by the problem of representation collapse in the latent space, which leads to low codebook utilization and limits the scalability of the codebook for large-scale training. Existing methods designed to mitigate representation collapse typically reduce the dimensionality of latent space at the expense of model capacity, which do not fully resolve the core issue. In this study, we conduct a theoretical analysis of representation collapse in VQ models and identify its primary cause as the disjoint optimization of the codebook, where only a small subset of code vectors are updated through gradient descent. To address this issue, we propose **SimVQ**, a novel method which reparameterizes the code vectors through a linear transformation layer based on a learnable latent basis. This transformation optimizes the *entire linear space* spanned by the codebook, rather than merely updating *the code vector* selected by the nearest-neighbor search in vanilla VQ models. Although it is commonly understood that the multiplication of two linear matrices is equivalent to applying a single linear layer, our approach works surprisingly well in resolving the collapse issue in VQ models with just one linear layer. We validate the efficacy of SimVQ through extensive experiments across various modalities, including image and audio data with different model architectures. The results show that SimVQ not only effectively addresses the problem of representation collapse but also proves highly adaptable and easy to implement, suggesting its broad applicability in diverse machine learning contexts.

## 1 INTRODUCTION

In recent years, vector quantization (VQ) (van den Oord et al., 2017; Razavi et al., 2019) has emerged as a foundational technique in unsupervised representation learning (Baevski et al., 2020; Bruce et al., 2024) and latent generative models (Rombach et al., 2022; Yu et al., 2022a;b; Borsos et al., 2023; Wang et al., 2023; Zhu et al., 2024b). By converting continuous representations into discrete codes, VQ models can effectively identify the inherent structure of data and enable various discrete modeling methods on continuous data, from high-quality image generation (Esser et al., 2021) to audio synthesis (Défossez et al., 2023). The recent success of Large Language Models (LLMs) (Achiam et al., 2023) has highlighted the effectiveness of next-token prediction as a powerful and versatile training objective. Consequently, VQ models are taken as the direct method to transform data from various modalities (Zhang et al., 2023a; Sun et al., 2024; Team, 2024) or scientific domains (Gao et al., 2024) to discrete sequences for next token prediction training. However, attempts to integrate VQ models as multimodal tokenizers to leverage the scaling laws of LLMs face significant challenges because of the difficulty of expanding the codebook. For example, the Chameleon model (Team, 2024) constrains its codebook size to $8k$, which is significantly trailing behind the vocabulary size of LLMs (e.g., LLaMA3's vocabulary size is $128k$ (Dubey et al., 2024)).

There is a broad agreement that increasing vocabulary size can consistently improve the performance of LLMs (Tao et al., 2024). However, recent studies (Zhu et al., 2024a) indicate that traditional VQ models often fail to utilize the additional parameters introduced by codebook expansion, leaving most codes inactive during training. The contradiction between codebook expansion and low codebook utilization in VQ models is known as the representation collapse problem (Roy et al., 2018),

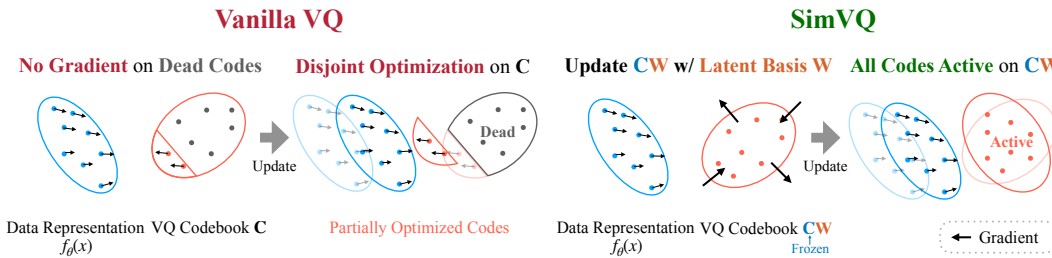

Figure 1: **Comparison of Vanilla VQ and SimVQ.** (a): (left) Disjoint optimization in Vanilla VQ. Only the nearest codes are updated, resulting in a high percentage of "dead" codes that are not updated. (b): (right) Joint optimization in SimVQ. The entire codebook is updated with a latent basis, ensuring all codes remain active.

where increasing the codebook size fails to improve the performance. To address these discrepancies, we conduct a theoretical analysis of the optimization procedure of VQ models and identify that the disjoint optimization of the codebook is the root cause of representation collapse. As illustrated in Fig. 1(a), the core mechanism of VQ models involves a nearest-neighbor replacement strategy, where the encoder's output features are replaced by the nearest vector in the codebook to serve as input to the decoder. The indices of the nearest vector are taken as the discrete representation of the data. This nearest-selection operator results in only a subset of codes being updated through gradient descent, while the remaining codes remain unchanged.

Recently, some approaches have been proposed to mitigate representation collapse. FSQ (Mentzer et al., 2024), LFQ (Yu et al., 2024) and ViTVQGAN (Yu et al., 2022a) reduce the dimension of the latent space to a very small scale (e.g., 8 vs. 128) to alleviate the curse of dimensionality, thereby improving the overlap between the encoder's features and the codebook. However, while these methods enhance codebook utilization, they do so at the cost of model capacity, leading to worse performance compared to vanilla VQ models when the codebook size is small and representation collapse is not severe. Another approach, VQGAN-LC (Zhu et al., 2024a), initializes the codebook with features extracted from the pre-trained CLIP model (Radford et al., 2021) to create a well-structured latent space that better matches the distribution of the encoder output. Nevertheless, the latent space defined by an external pre-trained model limits the model's ability to generalize to diverse datasets and reaches a performance plateau as the codebook size increases. These limitations highlight the need for a more effective method to improve codebook utilization without compromising model capacity or relying on external models.

We critically assess prevalent methodologies and reveal that optimizing the latent space rather than individual code vectors is key to preventing representation collapse. Building on this insight, we introduce a simple yet effective method, termed SimVQ, to directly update the latent space spanned by the codebook by linear transforming the code vectors via a learnable latent basis. Specifically, the vectors in the codebook are reparameterized as a linear combination of the basis in the learnable linear layer $W$:

$$C \in \mathbb{R}^{K \times d} \Rightarrow CW \text{ with } W \in \mathbb{R}^{d \times d}, \tag{1}$$

where $K$ denotes the codebook size and $d$ represents the dimension of latent space. This reparameterization with linear transformation disentangles the optimization of the codebook into two components: the coefficient matrix $C$ and the basis of linear space $W$ respectively. As illustrated in Fig. 1(b), by optimizing the basis matrix $W$, the latent space spanned by $CW$ is rotated and stretched to match encoder's output feature. The entire codebook is updated jointly to prevent the representation collapse problem. The simplicity of the proposed method makes it highly portable and easily adaptable for improving VQ-based models across a wide range of domains, requiring only *one linear layer*.

In summary, our contributions to vector quantized models are as follows:

- We theoretically analyze the representation collapse problem of VQ models and reveal that optimizing the latent space spanned by the codebook, rather than focusing on the individual code vectors, is crucial to addressing this issue.

- We propose a novel method, SimVQ, which reparameterizes the codebook vectors in VQ models via a linear transformation with a learnable latent basis. This simple yet effective approach is highly adaptable and easy to implement, making it broadly applicable across various machine learning contexts.

- We conduct an extensive evaluation of SimVQ across diverse modalities, including image and audio with different model architectures. The results show that SimVQ not only effectively addresses the representation collapse problem by achieving near-complete codebook utilization regardless of the codebook size, but also establishes new state-of-the-art performance. Furthermore, when scaling up the codebook size, SimVQ consistently delivers improved results.

## 2 RELATED WORK

VQ-VAE (van den Oord et al., 2017) is the pioneering work to encode data into discrete representations, which is further improved by VQ-VAE2 (Razavi et al., 2019) by employing a hierarchical architecture to enable richer representations. Building on these developments, VQGAN (Esser et al., 2021) combines VQ-VAE with adversarial networks to improve the perceptual quality of generated samples and establish a fundamental quantization protocol in latent generative models (Rombach et al., 2022; Yu et al., 2022b; Team, 2024). However, these methods suffer from a critical issue of representation collapse, as they struggle to scale the codebook size beyond 10k entries, limiting their scalability. In response to this challenge, several approaches have been proposed recently. DALLE (Ramesh et al., 2021) employs the gumbel-softmax trick (Jang et al., 2017) and stochastic sampling strategies to activate most codes during training. However, during inference, only a small subset of codes is utilized for quantization (Zhang et al., 2023b). Huh et al. (2023) proposes rescaling the vectors in the codebook during training to match the distributions in the latent space. VQGAN-FC (Yu et al., 2022a) introduces a method to map latent vectors into a lower-dimensional space followed by $l_2$ normalization to alleviate representation collapse. FSQ (Mentzer et al., 2024) extends this idea by projecting representations into a reduced-dimensional space, where they are quantized into a small set of fixed values. LFQ (Yu et al., 2024), a variant of FSQ, uses binary values for quantized representations, thereby simplifying the encoding process. While these methods improve the codebook utilization, they do so at the cost of model capacity by significantly reducing the dimensionality of latent space (often to as low as 8), leading to worse performance compared to vanilla VQ models when the codebook size is small and representation collapse is not severe. Additionally, VQGAN-LC (Zhu et al., 2024a) proposes to initialize the codebook using features extracted from the pre-trained CLIP model to avoid representation collapse. However, the reliance on the pre-trained model limits the VQ model's ability to generalize to diverse datasets and results in a performance plateau as the codebook size increases. In contrast, our method, SimVQ, effectively addresses the representation collapse problem with a simple linear layer, without sacrificing model capacity or relying on external pre-trained models.

## 3 REPRESENTATION COLLAPSE IN VQ MODELS

### 3.1 PRELIMINARIES

A vector quantized model is typically a reconstructive encoder-decoder architecture that includes a vector quantization layer to convert continuous representations into discrete codes. For simplicity, we represent an image with a single random variable $x$. Formally, the encoder $f_\theta$ maps the input image into a latent space, producing a continuous representation $z_e = f_\theta(x) \in \mathbb{R}^d$. This representation is then quantized using a learnable codebook $C = [q_1, \ldots, q_K] \in \mathbb{R}^{K \times d}$, where $q_i$ is a codebook vector. We define $\delta_k \in \{0, 1\}^{1 \times K}$ as a characteristic (one-hot) vector where only the $k$-th element is 1, such that $q_k = \delta_k C \in \mathbb{R}^{1 \times d}$. The quantization layer selects the nearest codebook vector $q_k$ by minimizing the Euclidean distance between $z_e$ and the codebook entries (van den Oord et al., 2017):

$$k = \arg\min_j \|z_e - q_j\|_2^2 = \arg\min_j \|z_e - \delta_j C\|_2^2. \tag{2}$$

The selected vector $q_k$ is then passed to the decoder $g_\phi$ to reconstruct the input image.

To enable gradient propagation through the non-differentiable characteristic vector $\delta_k$, the straight-through estimator (STE) (Bengio et al., 2013) is applied. During the backward pass, the gradient of $z_q = \delta_k C$ is copied to $z_e$ as follows,

$$z_q = \text{sg}(\delta_k C - z_e) + z_e, \quad \Rightarrow \frac{\partial z_q}{\partial z_e} = 1 \tag{3}$$

where $\text{sg}$ is the stop gradient operator, ensuring the gradient for $\delta_k C$ is discarded during the backward pass.

The learning objective is the combination of a reconstruction loss and commitment loss that ensures that the encoder commits to an embedding and the encoder's output does not drift:

$$\mathcal{L} = \log p(x|z_q) + \|\text{sg}(\delta_k C) - z_e\|_2^2 + \beta\|\delta_k C - \text{sg}(z_e)\|_2^2, \tag{4}$$

where $\log p(x|z_q)$ is typically the mean squared error (MSE) loss $\|x - g_\phi(z_q)\|_2^2$ for image and audio data.

## 3.2 DISJOINT OPTIMIZATION OF CODEBOOK

In VQ models, only the nearest code is selected and updated via gradient descent. Ideally, all codebook entries should be updated and utilized for decoding. However, experimental evidence shows that only a small fraction of the codebook gets updated and utilized, leading to what is known as the representation collapse problem (Roy et al., 2018). To investigate the root cause of this issue, we provide a theoretical analysis of the optimization dynamics in VQ models.

Due to the use of the straight-through estimator (STE) for gradient propagation, the codebook $C$ can only be updated through the gradient of the commitment loss, which is defined as:

$$\mathcal{L}_{commit}(C) = \|z_e - \delta_k C\|_2^2. \tag{5}$$

The codebook $C$ is updated according to the following equation, where $\eta$ is the learning rate:

$$C^{(t+1)} = C^{(t)} + \eta\mathbb{E}_{z_e}\left[\frac{\partial\mathcal{L}_{commit}(C^{(t)})}{\partial C^{(t)}}\right] = C^{(t)} - \eta\mathbb{E}_{z_e}\left[\delta_k^T\delta_k C^{(t)}\right] + \eta\mathbb{E}_{z_e}\left[\delta_k^T z_e\right] \tag{6}$$

where $\delta_k^T\delta_k$ is the Kronecker delta matrix, defined as:

$$(\delta_k^T\delta_k)_{ij} = \begin{cases} 1 & \text{if } i = j = k, \\ 0 & \text{otherwise.} \end{cases} \tag{7}$$

All vectors in $C$ will be updated and utilized if and only if the expectation $\mathbb{E}_{z_e}\left[\delta_k^T\delta_k\right]$ converges to the identity matrix. Unlike variational autoencoders (VAEs) (Kingma & Welling, 2013), which enforce a Gaussian distribution on the latent space via a KL-divergence penalty, VQ models optimize $z_e$ towards the selected codebook vectors $\mathbb{E}_{z_e}\left[\delta_k^T\delta_k C\right]$. At the same time, the selected codebook vectors are optimized towards the distribution of $z_e$, resulting in the same selected subset of vectors moving closer to $z_e$, somewhat akin to a cocoon effect. However, this disjoint optimization of the codebook leads to part of the codebook, specifically $(I - \mathbb{E}_{z_e}\left[\delta_k^T\delta_k\right])C$, remaining unupdated and underutilized once the optimization process begins. This phenomenon occurs because the optimization focuses only on a subset of codebook vectors, leaving other vectors effectively stagnant.

This analysis reveals the fundamental cause of representation collapse in VQ models: the disjoint optimization process that updates only a subset of codebook vectors. This insight forms the basis for our proposed solution, SimVQ, which aims to address this issue by optimizing the entire latent space spanned by the codebook, rather than individual code vectors.

## 4 ADDRESSING COLLAPSE WITH LATENT LINEAR TRANSFORMATION

### 4.1 REPARAMETERIZE CODES WITH LATENT BASIS

Let $W = \{w_1, \ldots, w_n\}$ be a basis of a linear space. Any vector $v$ in the space can be uniquely expressed as a linear combination of the basis vectors with coefficients $c_1, \ldots, c_n \in \mathbb{R}$:

$$v = c_1 w_1 + \cdots + c_n w_n = cW. \tag{8}$$

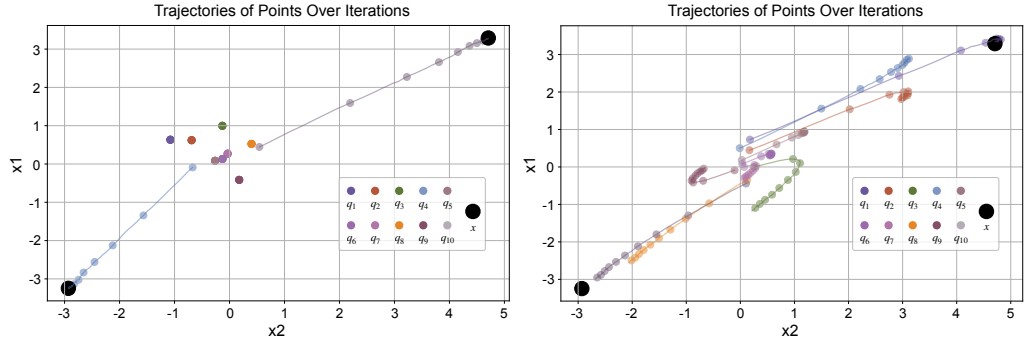

Figure 2: (a): (left) The optimization trajectory of the objective $\|\boldsymbol{x} - \boldsymbol{q}\|_2^2$, which is the same as vanilla VQ. Only a small fraction of points are updated while others remain inactive. (b): (right) The optimization trajectory of the objective $\|\boldsymbol{x} - \boldsymbol{q}\boldsymbol{w}\|_2^2$ with $\boldsymbol{q}$ frozen, which is the same as SimVQ. All the points are updated towards targets $x$.

Given the equivalence between $\boldsymbol{v}$ and $\boldsymbol{c}\boldsymbol{W}$ in the linear space, we can reparameterize each vector in the codebook of VQ models with a new basis matrix $\boldsymbol{W} \in \mathbb{R}^{d \times d}$. Specifically, the codebook $\boldsymbol{C} = \{\boldsymbol{c}_1, \ldots, \boldsymbol{c}_K\}$ can be reparameterized as:

$$\{\hat{\boldsymbol{c}}_1 \boldsymbol{W}, \ldots, \hat{\boldsymbol{c}}_N \boldsymbol{W}\} = \hat{\boldsymbol{C}} \boldsymbol{W} \in \mathbb{R}^{K \times d}. \tag{9}$$

This reparameterization introduces two components: the basis matrix $\boldsymbol{W}$ and the coefficient matrix $\hat{\boldsymbol{C}}$. In the following, we will discuss the optimization of both the basis matrix $\boldsymbol{W}$ and the coefficient matrix $\hat{\boldsymbol{C}}$. For simplicity, we slightly abuse $\boldsymbol{C}$ and $\hat{\boldsymbol{C}}$ below.

**Asymmetric Optimization Dynamics** While it is commonly accepted that multiplying two linear matrices is equivalent to a single linear layer, we argue that the disjoint optimization problem of the codebook in VQ models can be addressed by the basis transformation. In vanilla VQ models, only the codebook $\boldsymbol{C}$ is responsible for minimizing commitment loss, leading to the disjoint optimization problem where only the selected codes will be updated.

In contrast, when the codebook is reparameterized as $\boldsymbol{C}\boldsymbol{W}$, both the basis $\boldsymbol{W}$ and the coefficient matrix $\boldsymbol{C}$ contribute to minimizing the commitment loss. The gradients $\frac{\partial \mathcal{L}}{\partial \boldsymbol{W}}$ and $\frac{\partial \mathcal{L}}{\partial \boldsymbol{C}}$ can simultaneously reduce the loss. As a result, the optimization of the reparameterized codebook can be divided into three scenarios:

- Updating $\boldsymbol{C}$ with $\boldsymbol{W}$ frozen: Only the **selected** codes adapt to the latent distribution of $z_e$, as depicted on Fig. 1(a). The vanilla VQ is a special case of this scenario with $\boldsymbol{W} = \boldsymbol{I}$.
- Updating $\boldsymbol{W}$ with $\boldsymbol{C}$ frozen: The **entire** codebook $\boldsymbol{C}\boldsymbol{W}$ adjusts to the latent distribution of $z_e$. The basis matrix $\boldsymbol{W}$ rotates and stretches the codebook space as shown on Fig. 1(b).
- Updating both $\boldsymbol{C}$ and $\boldsymbol{W}$: The selected subset of codes moves towards $z_e$ while the space spanned by $\boldsymbol{W}$ undergoes simultaneous rotation and stretching.

## 4.2 TOY EXAMPLES

To highlight the difference in optimization between $\boldsymbol{C}$ and $\boldsymbol{C}\boldsymbol{W}$, we conduct a toy experiment in a two-dimensional setting and visualize the optimization process in Fig. 2 and Fig. 3. We randomly sample two target points $\boldsymbol{x}$ from Gaussian distribution as follows:

$$\boldsymbol{x}_1 \sim \mathcal{N}(\begin{pmatrix} 2 \\ 2 \end{pmatrix}, \begin{pmatrix} 1 & 0 \\ 0 & 1 \end{pmatrix}), \quad \boldsymbol{x}_2 \sim \mathcal{N}(\begin{pmatrix} -2 \\ -2 \end{pmatrix}, \begin{pmatrix} 1 & 0 \\ 0 & 1 \end{pmatrix}). \tag{10}$$

Then we initialize 10 learnable vectors $\boldsymbol{q}$ from a Gaussian distribution:

$$\{\boldsymbol{q}_i\}_{i=1}^{10} \sim \mathcal{N}(\begin{pmatrix} 0 \\ 0 \end{pmatrix}, \begin{pmatrix} 1 & 0 \\ 0 & 1 \end{pmatrix}), \tag{11}$$

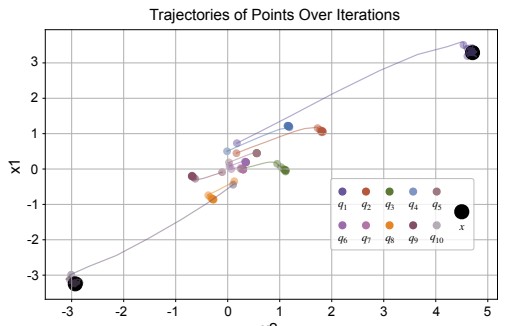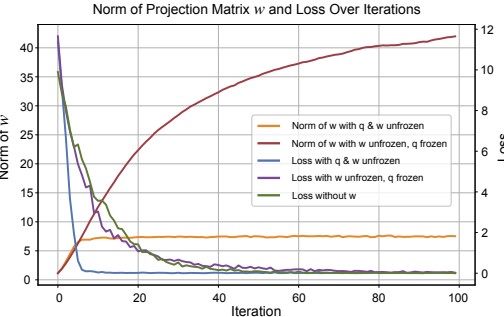

Figure 3: (a): (left) The optimization trajectory of the optimization objective: $\|\boldsymbol{x} - \boldsymbol{qw}\|_2^2$ with both $\boldsymbol{q}$ and $\boldsymbol{w}$ unfrozen. (b): (right) The Frobenius norm of the projection matrix $\boldsymbol{w}$ and loss curves. The loss quickly converges to 0 with $\boldsymbol{w}$ almost unchanged.

---

**Algorithm 1** Training Procedure for SimVQ

**Input:** Encoder $f_\theta$, Decoder $g_\phi$, Codebook $\boldsymbol{C} \in \mathbb{R}^{K \times d}$, Linear projector matrix $\boldsymbol{W}_\psi$, commitment weight $\beta$.
**Output:** Model parameters $\theta, \phi, \psi$ and Codebook $\boldsymbol{C}$.
Initialize Codebook $\boldsymbol{C}$ with Gaussian distribution and freeze the parameter of Codebook;
**repeat**
    Draw $x \sim p_{data}(\boldsymbol{x})$;
    $z_e = f_\theta(x)$;
    /* Replace $q_j$ in vanilla VQ with proposed $q_j \boldsymbol{W}_\psi$.
    Nearest code search: $k = \arg\min_j \|z_e - q_j \boldsymbol{W}_\psi\|_2^2$, where $q_j \in \boldsymbol{C}$;
    Straight Through Estimation: $z_q = \text{sg}(q_k \boldsymbol{W}_\psi - z_e) + z_e$;
    $\hat{x} = g_\phi(z_q)$;
    Minimize $\mathcal{L}(\theta, \phi, \psi) = \text{MSE}(x, \hat{x}) + \beta\|z_e - \text{sg}(q_k \boldsymbol{W}_\psi)\|_2^2 + \|\text{sg}(z_e) - q_k \boldsymbol{W}_\psi\|_2^2$;
**until** converged

---

During training with gradient descent, we introduce perturbation noise $\mathcal{N}(0, 0.01)$ to the targets. In Fig. 2(a), the optimization objective is similar to vanilla VQ: $\|\boldsymbol{x} - \boldsymbol{q}\|_2^2$. Only the nearest points $\boldsymbol{q}_4$ and $\boldsymbol{q}_{10}$ are updated. In contrast, in Fig. 2(b), the optimization objective $\|\boldsymbol{x} - \boldsymbol{qw}\|_2^2$ is similar to SimVQ with the points reparameterized by a learnable latent basis $\boldsymbol{w}$ and $\boldsymbol{q}$ frozen, resulting in the entire codebook $\{\boldsymbol{q}\}_{i=1}^{10}$ being *jointly* updated.

*Remark* 4.1. The simultaneous optimization of the latent basis $\boldsymbol{w}$ and the coefficient matrix $\boldsymbol{q}$ may lead to the collapse.

We provide an example in Fig. 3(a) where the optimization objective is $\|\boldsymbol{x} - \boldsymbol{qw}\|_2^2$ with $\boldsymbol{q}$ unfrozen this time. In the training process, only the nearest point $\boldsymbol{q}_1$ and point $\boldsymbol{q}_{10}$ move towards the target point, while other points remain almost unchanged. We also visualize the loss curve in Fig. 3(b). The optimization objective with both $\boldsymbol{q}$ and $\boldsymbol{w}$ unfrozen converges quickly, where the norm of basis $\boldsymbol{w}$ is much smaller than the objective with $\boldsymbol{q}$ frozen. This indicates that the disjoint optimization of the codebook persists: $\boldsymbol{q}$ can directly commit to the loss and dominate the optimization process, with $\boldsymbol{w}$ being largely ignored, leading to the collapse quickly.

### 4.3 JOINT OPTIMIZATION OF THE CODEBOOK

We propose SimVQ by simply using a learnable basis $\boldsymbol{W} \in \mathbb{R}^{d \times d}$ to reparameterize the codebook such that the codebook is transformed into $\boldsymbol{CW}$. The pseudo-code for this approach is provided in Algorithm 1. During training, we optimize only the latent basis matrix $\boldsymbol{W}$, while keeping the coefficient matrix $\boldsymbol{C}$ frozen. The commitment loss for SimVQ is defined as:

$$\mathcal{L}_{commit}(z_e, q_k) = \|z_e - \delta_k \boldsymbol{CW}\|_2^2. \tag{12}$$

The vanilla VQ model is a special case of SimVQ, where the linear basis matrix $\boldsymbol{W}$ is fixed as the identity matrix $\boldsymbol{I}$. The update for $\boldsymbol{W}$ with learning rate $\eta$ is:

$$\boldsymbol{W}^{(t+1)} = \boldsymbol{W}^{(t)} - \eta\frac{\partial\mathcal{L}_{commit}(z_e, \boldsymbol{q}_k)}{\partial\boldsymbol{W}^{(t)}} = (\boldsymbol{I} - \eta\mathbb{E}_{z_e}\left[\boldsymbol{C}^T\delta_k^T\delta_k\boldsymbol{C}\right])\boldsymbol{W}^{(t)} + \eta\mathbb{E}_{z_e}\left[\boldsymbol{C}^T\delta_k^T z_e\right]. \quad (13)$$

The term $\mathbb{E}\left[\boldsymbol{C}^T\delta_k^T\delta_k\boldsymbol{C}\right]$ represents the expectation of the quadratic form, and simplifies to $\mathbb{E}[\boldsymbol{q}_k^T\boldsymbol{q}_k]$. Since the codes are randomly sampled from a Gaussian distribution, we have:

$$\mathbb{E}\left[\boldsymbol{q}_k^T\boldsymbol{q}_k\right] = \boldsymbol{I}, \text{where } \boldsymbol{q} \sim \mathcal{N}(0, 1), \quad (14)$$

which ensures that all elements of $\boldsymbol{W}$ are updated. As training progresses, the latent basis $\boldsymbol{W}$ converges to:

$$\lim_{t\to\infty}\boldsymbol{W}^{(t)} = \mathbb{E}_{z_e}\left[\boldsymbol{q}_k^T z_e\right] \quad (15)$$

Thus, in the limit:

$$\lim_{t\to\infty}\boldsymbol{q}_k\boldsymbol{W}^{(t)} = \mathbb{E}\left[\boldsymbol{q}_k\boldsymbol{q}_k^T e\right] = \mathbb{E}\left[e\right] \quad (16)$$

At convergence, the product $\boldsymbol{q}_k\boldsymbol{W}$ equals the nearest feature.

### 4.4 Efficiency Analysis

SimVQ demonstrates greater efficiency than vanilla VQ due to its asymmetric training strategy, wherein the codebook $\boldsymbol{C}$ remains static and only the linear projection $\boldsymbol{W}$ is optimized. This approach results in a significant reduction in memory usage during the gradient backpropagation process. In vanilla VQ, the memory cost for the optimization of the codebook is $O(Kd)$, where $K$ is the number of vectors in the codebook, and $d$ is the dimension of each vector. In our experiments, $K = 65,536$ is much larger than $d = 128$. As the vocabulary size increases, the memory required for backpropagation grows proportionally, significantly impacting resource consumption. In contrast, SimVQ's memory cost for backpropagation is only $O(d^2)$ because the codebook $\boldsymbol{C}$ is fixed, and only the linear layer $\boldsymbol{W}$ is updated. This results in a constant memory requirement in backpropagation, independent of the vocabulary size. The $d \times d$ scaling becomes particularly advantageous as $K$ increases in practical applications. This structural design minimizes the computational overhead and improves training efficiency, especially when dealing with large vocabularies.

## 5 Experiments

To assess the efficacy and versatility of the proposed SimVQ, we conduct experiments across both image and audio modalities. Subsequently, we analyze the learned linear layer to investigate the latent basis. The experimental configurations are listed in Appendix A.1.

### 5.1 Vision Modality

#### 5.1.1 Implementation Details

To rigorously evaluate the proposed SimVQ, we reproduce all the VQ models listed in Tab. 1 using the same architecture of VQGAN (Esser et al., 2021) with the quantization layer different only. Among the baselines, for VQGAN-FC (Yu et al., 2022a), we follow the original setting to reduce the dimension of the latent space to $8$ followed by $l_2$ normalization to improve codebook utilization. For FSQ (Mentzer et al., 2024), we adopt a codebook size of $[8, 8, 8, 5, 5, 5,]$ as recommended, to approximately match the default codebook size. For VQGAN-LC (Zhu et al., 2024a), we follow them and leverage an external pre-trained CLIP model to extract features of the training dataset in advance for a well-defined latent space. All models are trained on the ImageNet (Deng et al., 2009) dataset for 50 epochs with a batch size of 256. Input images are processed at a resolution of $128 \times 128$ pixels and downsampled by a factor of 8, yielding a feature map of $16 \times 16 \times 128$, where 128 is the dimension of the latent space. We set the default codebook size to a large number of $2^{16} = 65536$ rather than the traditional number 8192 to highlight the representation collapse problem. Performance is evaluated using rFID, LPIPS, PSNR, and SSIM metrics on the ImageNet validation set.

Table 1: Reconstruction performance on ImageNet-1k with a resolution of $128 \times 128$. All models are trained using images downsampled into $16 \times 16$ tokens. † Results are reproduced using the codebook size of $[8, 8, 8, 5, 5, 5]$ to approximately match $65,536$. + Following VQGAN-LC, we extract CLIP features with the codebook frozen.

| Method | Latent dim | Codebook size | Util↑ | rFID↓ | LPIPS↓ | PSNR↑ | SSIM↑ |
|---|---|---|---|---|---|---|---|
| VQGAN (Esser et al., 2021) | 128 | 65,536 | 1.4% | 3.74 | 0.17 | 22.20 | 70.6 |
| VQGAN-EMA (Razavi et al., 2019) | 128 | 65,536 | 4.5% | 3.23 | 0.15 | 22.89 | 72.3 |
| VQGAN-FC (Yu et al., 2022a) | 128 | 65,536 | 1.4% | 5.33 | 0.18 | 21.45 | 68.8 |
| VQGAN-FC (Yu et al., 2022a) | 8 | 65,536 | 100.0% | 2.63 | 0.13 | 23.79 | 77.5 |
| FSQ† (Mentzer et al., 2024) | 16 | 64,000 | 100.0% | 2.80 | 0.13 | 23.63 | 75.8 |
| LFQ (Yu et al., 2024) | 6 | 65,536 | 100.0% | 2.88 | 0.13 | 23.60 | 77.2 |
| VQGAN-LC-CLIP+ (Zhu et al., 2024a) | 768 | 65,536 | 100.0% | 2.40 | 0.13 | 23.98 | 77.3 |
| SimVQ (ours) | 128 | 65,536 | 100.0% | **2.24** | **0.12** | **24.15** | **78.4** |
| SimVQ (ours) | 128 | 262,144 | 100.0% | **1.99** | **0.11** | **24.68** | **80.3** |

### 5.1.2 MAIN RESULTS

Tab. 1 presents the reconstruction performance of various VQ models on image data. We make three key observations: 1) Traditional VQGAN models utilize only a very small subset of the codebook, with a utilization rate of just $1.4\%$. Although VQGAN-EMA is proposed to improve codebook utilization, especially when the codebook size scales up to $65k$, it still suffers from severe representation collapse. 2) Recently proposed methods, such as LFQ, FSQ, and VQGAN-FC, effectively improve codebook utilization to $100\%$. However, these methods require reducing the latent space to a very low dimension. For example, applying VQGAN-FC to the standard latent dimension of 128 results in severe representation collapse and degraded reconstruction performance. Additionally, these models face limitations in model capacity due to the low-dimensional latent space. While they achieve full codebook utilization, their reconstruction quality on rFID score lags significantly behind SimVQ. 3) VQGAN-LC-CLIP leverages an external pre-trained CLIP model to provide a well-defined latent space. However, VQGAN-LC relies on CLIP features pre-trained on much larger datasets than ImageNet, which introduces generalization issues and a lower performance ceiling (degradation issue in Tab. 2). In contrast, SimVQ can be applied to a wide range of data types and achieves superior performance (rFID 2.40 vs. 2.24) without the limitations imposed by a pre-trained feature extraction model.

### 5.1.3 ABLATION STUDY

**Codebook Size** In Tab. 2, we explore the impact of different codebook sizes, ranging from $1k$ to $262k$, which is the level of LLM's vocabulary size. SimVQ consistently improves performance as the codebook size increases. For instance, the rFID score decreases to 1.99, and SSIM surpasses 80.0. In contrast, VQGAN-LC-CLIP encounters performance degradation, with the rFID score worsening from 2.62 to 2.66 when the codebook size is increased from $100,000$ to $200,000$.

**Codebook Optimization Strategy** We investigate codebook initialization and the training of the linear layer in Tab. 3. Our findings are as follows: 1) The codebook is robust to different initialization strategies, yielding similar results with both Gaussian and uniform initialization. 2) When the codebook is updated during training, SimVQ continues to address the representation collapse issue, though with a slight degradation in performance.

### 5.2 AUDIO MODALITY

### 5.2.1 IMPLEMENTATION DETAILS

We use LibriTTS dataset (Zen et al., 2019) for audio-based VQ model training. The baselines such as Encodec (Défossez et al., 2023), Vocos (Siuzdak, 2024), and SpeechTokenizer (Zhang et al., 2024) are based on residual vector quantization. Our SimVQ model adopts the same architecture as WavTokenizer (Ji et al., 2024) with the only modification being the replacement of their EMA codebook with our one linear layer reparameterization method. We train SimVQ on LibriTTS-580h

Table 2: Ablation study on the effect of various codebook sizes on ImageNet at a resolution of $128 \times 128$. † We directly copy the reported results of VQGAN-LC from the original paper on ImageNet $256 \times 256$ resolution.

| Method | Codebook Size | Util↑ | rFID↓ | LPIPS↓ | PSNR↑ | SSIM↑ |
|---|---|---|---|---|---|---|
| VQGAN-LC-CLIP† | 50,000 | 99.9% | 2.75 | 0.13 | 23.8 | 58.4 |
| VQGAN-LC-CLIP† | 100,000 | 99.9% | 2.62 | 0.12 | 23.8 | 58.9 |
| VQGAN-LC-CLIP† | 200,000 | 99.8% | 2.66 | 0.12 | 23.9 | 59.2 |
| SimVQ | 1,024 | 100.0% | 3.67 | 0.16 | 22.34 | 70.8 |
| SimVQ | 8,192 | 100.0% | 2.98 | 0.14 | 23.23 | 74.7 |
| SimVQ | 65,536 | 100.0% | 2.24 | 0.12 | 24.15 | 78.4 |
| SimVQ | 262,144 | 100.0% | **1.99** | **0.11** | **24.68** | **80.3** |

Table 3: Ablation study of codebook optimization.

| Initialization | Trainable | Util↑ | rFID↓ | LPIPS↓ | PSNR↑ | SSIM↑ |
|---|---|---|---|---|---|---|
| Gaussian | Yes | 100.0% | 2.31 | 0.12 | 24.04 | 77.2 |
| Uniform | No | 100.0% | 2.31 | 0.12 | 24.15 | 78.4 |
| Gaussian | No | 100.0% | **2.24** | **0.12** | **24.15** | **78.4** |

for 50 epochs with a batch size of 64. Note that WavTokenizer is trained with a 3-second window size for optimal performance, we train SimVQ using a 1-second window to accelerate training. For objective evaluation of the reconstructed audio, we follow Vocos (Siuzdak, 2024) and employ metrics such as UTMOS (Saeki et al., 2022), PESQ (Rix et al., 2001), STOI, and the F1 score for voiced/unvoiced classification (V/UV F1). UTMOS is particularly valuable as it produces scores highly correlated with human evaluations.

### 5.2.2 MAIN RESULTS

Tab. 4 presents the reconstruction performance of various VQ models on audio data. Baseline models using residual vector quantization perform significantly worse than SimVQ, even when utilizing much larger bandwidths. Despite using the same architecture as WavTokenizer, our model, which replaces the quantization layer with SimVQ, achieves superior performance with a 1-second window size and maintains nearly 100% codebook utilization when scaling up to a size of 262,144. The consistent performance of the SimVQ model across both image and audio data demonstrates that SimVQ is a general method for addressing the representation collapse problem in VQ models and can be effectively applied across multiple modalities.

### 5.3 ANALYSIS

In Fig. 4(a), we plot the rank of the latent basis matrix over training epochs. Notably, SimVQ demonstrates the ability to adaptively adjust the dimensionality of the latent space. Specifically, when the codebook size is increased from $65,536$ to $262,144$, the rank of the latent basis matrix decreases more rapidly and converges to a lower value. This observation suggests that a larger codebook can effectively alleviate the pressure on the latent space dimensionality, allowing the model to represent data more efficiently. Additionally, despite the rank decreasing to a lower-dimensional space, SimVQ maintains 100% codebook utilization, highlighting its superiority over VQGAN-FC, which struggles when increasing the latent dimension from 8 to 128. We also calculate the Frobenius norm of the latent basis matrix, as shown in Fig. 4. The norm of a codebook size of $262,144$ is slightly large than for $65,536$, indicating that a larger codebook can span a broader area in the linear space. For a comprehensive evaluation, we also provide the reconstruction loss curve on the

Table 4: Reconstruction performance on LibriTTS test-clean/test-other dataset. ∗ WavTokenizer is trained with a window size of 3 seconds.The bandwidth of 0.9kbps, 0.975kbps, 1.2kbps, 1.35kbps means the codebook size of 4096, 8192, 65536, 262144 respectively.

| Method | Bandwidth | Util↑ | UTMOS↑ | PESQ↑ | STOI↑ | V/UV F1↑ |
|---|---|---|---|---|---|---|
| GT | - | - | 4.06/3.48 | - | - | - |
| EnCodec (Défossez et al., 2023) | 3.0kbps | - | 2.31/2.09 | 2.05/2.05 | 0.90/0.88 | 0.92/0.89 |
| Vocos (Siuzdak, 2024) | 3.0kbps | - | 3.53/3.06 | 2.40/2.19 | 0.92/0.90 | 0.94/0.91 |
| SpeechTokenizer (Zhang et al., 2024) | 3.0kbps | - | 3.56/3.02 | 1.93/1.74 | 0.88/0.84 | 0.93/0.89 |
| WavTokenizer (Ji et al., 2024) | 0.9kbps | 100/100% | 3.74/3.43* | 2.01/2.26* | 0.89/0.89* | 0.92/0.92* |
| SimVQ (ours) | 0.9kbps | 100.0/100.0% | **4.00/3.51** | **2.33**/2.08 | **0.91**/0.88 | **0.94**/0.91 |
| WavTokenizer (Ji et al., 2024) | 0.975kbps | 68/-% | 4.02*/- | 2.39*/- | 0.92*/- | 0.94*/- |
| WavTokenizer (Ji et al., 2024) | 1.05kbps | 27/-% | 4.00*/- | 2.36*/- | 0.81*/- | 0.94*/- |
| SimVQ (ours) | 0.975kbps | 99.4/99.4% | 4.03/3.52 | 2.42/2.15 | 0.92/0.88 | 0.94/0.92 |
| SimVQ (ours) | 1.2kbps | 99.4/99.0% | 4.03/3.52 | 2.54/2.26 | 0.93/0.90 | 0.94/0.92 |
| SimVQ (ours) | 1.35kbps | 95.6/94.7% | **4.03/3.53** | **2.61/2.31** | **0.93/0.90** | **0.95/0.93** |

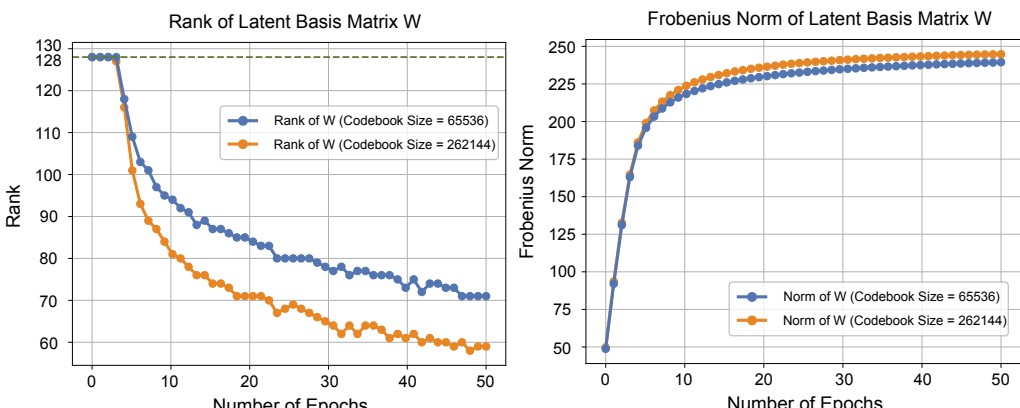

Figure 4: (a):(left) The rank of latent basis matrix $W$ over training epochs. (b):(right) The Frobenius norm of latent basis matrix $W$ over training epochs.

ImageNet validation dataset in Appendix A.2. The results consistently show that SimVQ achieves improved performance, further validating the effectiveness of our approach.

## 5.4 QUALITATIVE EVALUATION

We qualitatively compare the reconstruction quality of both images and audio in Appendix A.3. SimVQ achieves better reconstruction quality with an enlarged codebook size. For images, SimVQ with a larger codebook effectively preserves fine details, such as "eyes" and "text," which are challenging for vanilla VQ models. For audio, SimVQ retains more acoustic details in both spectrograms and waveforms, as demonstrated in Fig. 7 and Fig. 8.

## 6 CONCLUSION

In this paper, we explore the representation collapse problem in VQ models. We conduct a theoretical analysis of the optimization process in VQ models and propose a simple yet effective method, SimVQ, to address this issue. Our method addresses the representation collapse by jointly optimizing the latent space through linear transformation with one linear layer. Experimental results demonstrate that SimVQ outperforms previous approaches on both image and audio datasets, highlighting its broad applicability across diverse machine learning tasks.

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

# A APPENDIX

## A.1 EXPERIMENTAL CONFIGURATIONS

Table 5: Experimental configurations on image and audio.

| Config | Image | Audio |
|---|---|---|
| inputs | pixels | window size |
| input size | $128 \times 128 \times 3$ | $24,000 \times 1$ |
| batch size | 256 | 64 |
| training epochs | 50 | 50 |
| quantized sequence length | $16 \times 16$ | 75 |
| **optimization** | | |
| optimizer | AdamW | AdamW |
| learning rate | 1e-4 | 1e-4 |
| learning rate schedule | constant | constant |
| warmup epochs | 0 | 0 |
| commitment coefficient | 1.0 | 1000.0 |
| adversarial coefficient | 0.1 | 1.0 |
| **data augmentations** | | |
| random horizontal flip | true | false |

## A.2 LOSS CURVE

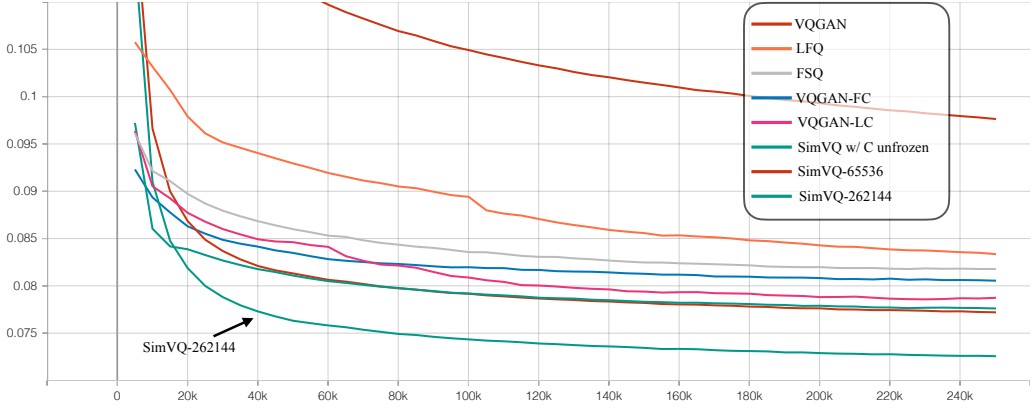

Figure 5: The loss curve over epochs of different models on the validation dataset.

## A.3 QUALITATIVE CASES

| Origin | vanilla VQ 65,536 | SimVQ 1,024 | SimVQ 8,192 | SimVQ 65,536 | SimVQ 262,144 |

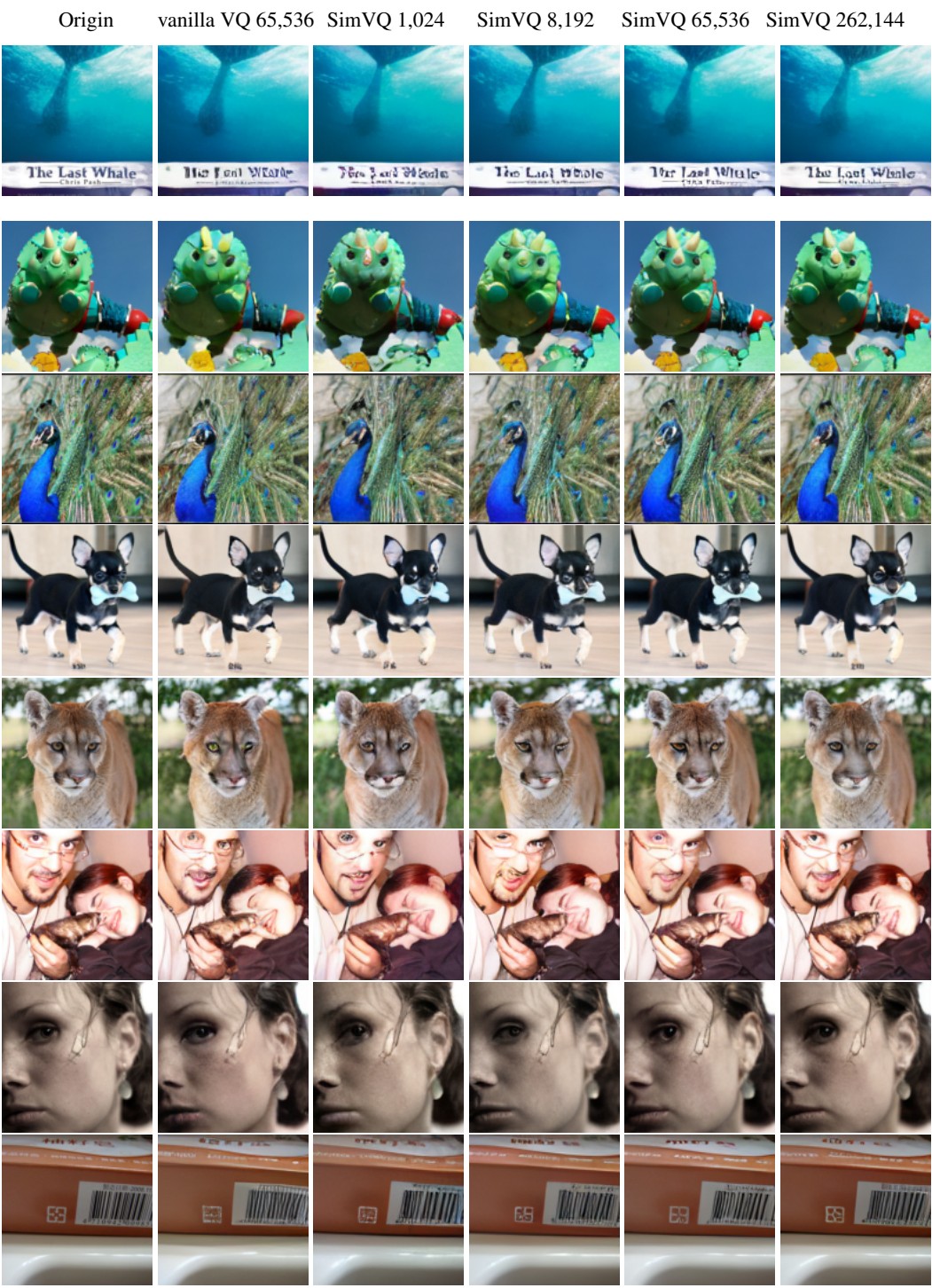

Figure 6: Image reconstruction samples with different codebook sizes.

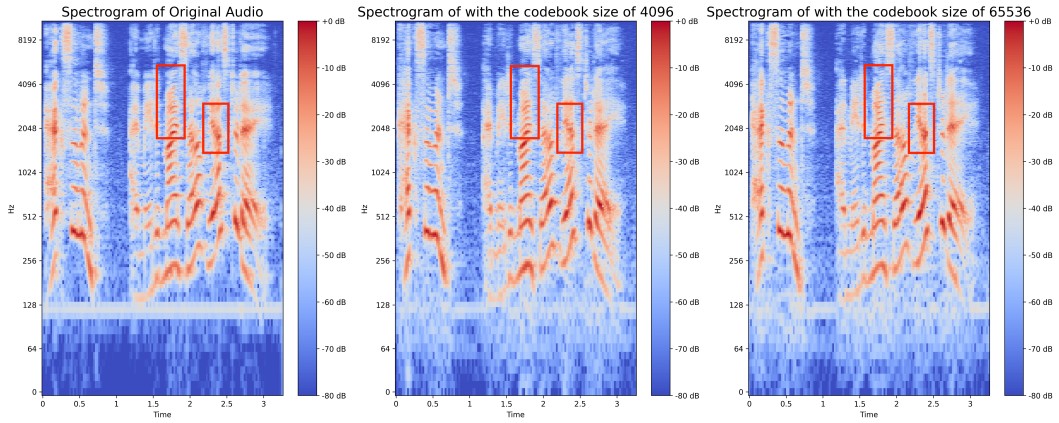

Figure 7: The spectrogram of audio reconstruction samples with different codebook sizes.

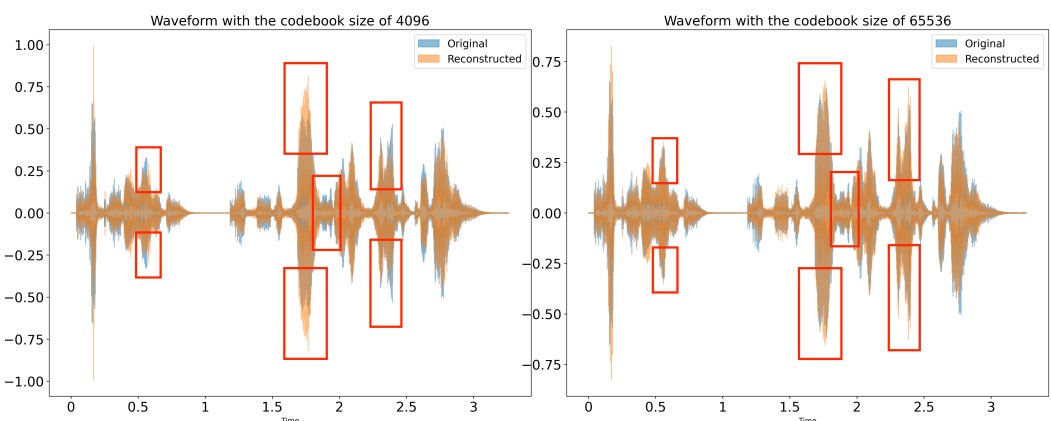

Figure 8: The waveform of audio reconstruction samples with different codebook sizes.

