# OpenReview forum: "Addressing Representation Collapse in Vector Quantized Models with One Linear Layer"
_ICLR.cc/2025/Conference — ICLR 2025 Conference Withdrawn Submission_

### Official Review · Reviewer_GEcJ · 2024-10-26

**Soundness:** 2
**Presentation:** 2
**Contribution:** 1
**Rating:** 3
**Confidence:** 5

**Summary:**

This paper addresses the issue of representation collapse in Vector Quantization (VQ) models, which hampers codebook utilization and limits scalability during large-scale training. Representation collapse occurs because the optimization process updates only a small subset of code vectors, leading to inefficient use of the latent space. Existing solutions reduce the dimensionality of the latent space but at the cost of model capacity.
To solve this, the authors propose SimVQ, a novel method that reparameterizes code vectors through a linear transformation based on a learnable latent basis. This approach optimizes the entire linear space spanned by the codebook, rather than just updating selected code vectors as in traditional VQ models.

While showing good reconstruction results, there are some shortcomings for this work.

**Strengths:**

1. A simple linear layer can be used to address the codebook collapse problem and obtain better reconstruction results.
2. Experiments were demonstrated on two modalities.

**Weaknesses:**

1. The idea using a linear combination of "atoms" and "coefficients" in the latent space has been explored on existing works.
    1) Auto-regressive Image Generation Using Residual Quantization.
    2) SC-VAE: Sparse Coding-based Variational Autoencoder with Learned ISTA.
    No discussion about these relevant works on the related work section. No comparisons with these methods were shown on the experiments section.
2. Though showing better reconstruction results. A very important aspect of vq-vae based model is to generate high-quality data (image, video, audio).  No generation results were exhibited. If I understand correctly, the proposed model can not discrete the latent representations. The latent representations were transformed into coefficients of codebook. How will you train a transformer model based on these coefficients?

**Questions:**

The work missed the generation results, which is a very important component of the proposed model. I would recommend the authors include generation results.

---

### Official Review · Reviewer_q9WW · 2024-10-30

**Soundness:** 2
**Presentation:** 3
**Contribution:** 2
**Rating:** 5
**Confidence:** 4

**Summary:**

The author proposes a simple way to solve the “representation collapse” problem in VQVAE. In VQVAE, the latent vector is being quantized to a one-hot vector with stop-gradient operator. Only one element of the dictionary is updated during training. The updated dictionary will likely to attract more latent vector which reinforce that dictionary element to be more activate. This results in only a subset off dictionary used in VQVAE is not utilized after training. The author proposes a simple way to solve this problem. They parametrized the dictionary C = C_hat W, where C_hat is a random matrix, and W is trainable parameter. In this case, all dictionary element must be updated during each backward pass.

**Strengths:**

1. The idea is simple! I really like it.
2. It solve a problem that's annoying, important but also relatively ignored by the community.

**Weaknesses:**

1. The method can be understood as a low rank re-parametrization. I felt like this method should have lots of connection to classic dictionary learning, sparse coding, and vector quantization. But this connection is not explored in the paper. I'm not that familiar with either the current VQVAE or dictionary learning literature. But I know this idea of "dead unit" is not new problem and people must tried something. Maybe other reviewers/AC can also help if they're familiar with these literatures.
2. The author claim other method that proposes to solve the similar problem require the dictionary element's dimension to be small. Is this true? I happen to read [1]. Correct me if I'm wrong but I don't think they has this assumption. And the author didn't compare their method to this method. Just to be clear, I don't mean to ask the author to benchmark against all the method. I think this is still very simple and cool method. But I simply don't really understand the logic behind WHY other method assume dictionary element's dimension to be small.
3. I personally think claiming "theoretically analyze" the dead unit phenomenon is too much. The analysis mentioned in the paper is not rigorous. To do so, I think you want to prove or cite some result for single layer dictionary learning.
4.By the way the index for equation 9 should be cK not cN?


[1] M, Huh, et al. Straightening Out the Straight-Through Estimator: Overcoming Optimization Challenges in Vector Quantized Networks

**Questions:**

1. How does the method works for just simple vector quantization problem (1 sparse sparse coding or clustering)? Would the same logic apply?

---

### Official Review · Reviewer_SoK8 · 2024-11-02

**Soundness:** 2
**Presentation:** 2
**Contribution:** 2
**Rating:** 5
**Confidence:** 4

**Summary:**

The paper argues that a fundamental cause of representation collapse in VQ models stems from the disjoint optimization process, where only a subset of codebook vectors is updated. To address this problem, the authors propose reparameterizing the code vectors through linear transformation layer based on a learnable latent basis. This transformation allows optimization across the entire linear space spanned by the codebook, rather than only updating the single selected code vectors. Experiments on image and audio reconstruction tasks with various model architectures demonstrate that this approach outperforms the baselines.

**Strengths:**

The paper is  easy to follow.

The proposed codebook reparameterization, which ensures updates across the entire codebook, is a promising approach.

The solution is straightforward yet effectively addresses codebook collapse, and the experiments on reconstruction tasks are convincing.

**Weaknesses:**

**W1:** Limited Literature Review: The paper lacks a comprehensive review of related work, omitting several relevant methods that tackle codebook collapse, such as SQ-VAE [1], VQ-WAE [2], HVQ-VAE [3], and CVQ-VAE [4].  I find that the insight from Section 3.2 isn’t particularly novel. Codebook collapse is often attributed to the initialization process, where latents tend to concentrate around a few codebook vectors. During training, latents can easily overfit to these vectors (referred to as the disjoint optimization process in this paper), leaving the remaining code vectors unused. Simple techniques like codebook reset [5], which randomly reinitializes unused or infrequently used codewords to one of the encoder outputs or to vectors near frequently used codewords, can help mitigate this issue.

&nbsp;

**W2:** Missing Generation Results: Since VQ-VAE aims to learn meaningful discrete representations for downstream tasks, it would be valuable to see generation results to verify the quality of the codebook. With the new codewords represented as linear combinations of W, it is unclear how these representations will impact codeword effectiveness. Generation results could clarify this.

&nbsp;

**W3:** Missing Baselines: Important baseline methods such as SQ-VAE [1], VQ-WAE [2], HVQ-VAE [3], and CVQ-VAE [4] are not included in the experimental comparisons.

&nbsp;


[1] Takida, Yuhta, et al. "Sq-vae: Variational bayes on discrete representation with self-annealed stochastic quantization." ICML (2022).

[2] Vuong, Tung-Long, et al. "Vector quantized wasserstein auto-encoder." ICML (2023).

[3] Williams, Will, et al. "Hierarchical quantized autoencoders." Neurips (2020)

[4] Zheng, Chuanxia, and Andrea Vedaldi. "Online clustered codebook." Proceedings of the IEEE/CVF International Conference on Computer Vision. 2023.

[5] Dhariwal, Prafulla, et al. "Jukebox: A generative model for music." arXiv preprint arXiv:2005.00341 (2020).

**Questions:**

Questions on Codebook Reparameterization: The new codebook is represented as CW, raising several questions:

- "Remark 4.1. The simultaneous optimization of the latent basis w and the coefficient matrix q may lead to the collapse.".  Why is this the case?

- It is unclear whether the analysis of codebook utilization in this paper is based on C or CW, as the codebook initialization ablation appears to be applied to C.

---

### Official Review · Reviewer_AQdz · 2024-11-05

**Soundness:** 2
**Presentation:** 3
**Contribution:** 2
**Rating:** 3
**Confidence:** 3

**Summary:**

The paper titled "Addressing Representation Collapse in Vector Quantized Models with One Linear Layer" describes an apporach which addresses the representation collapse problem in VQ where only a small subset of codebook vectors are effectively utilized, primarily due to disjoint optimization during training. The paper introduces SimVQ, which addresses this by reparameterizing code vectors through a learnable linear transformation layer that optimizes the entire linear space spanned by the codebook, rather than just updating individual selected code vectors.

**Strengths:**

1. The authors identify disjoint optimization of the codebook as the key reason for representation collapse in VQ
2. The proposed approach SimVQ only requires adding one linear layer for implementation
3. Maintains effectiveness with increasing codebook size

**Weaknesses:**

1. the main contribution of the paper can be summed up by the fact that if assigning one codeword to the encoder output causes representation collapse, one can use a weighted combination of the whole codebook. While this may work well in practise, I dont think a whole new paper is needed on this.
2. toy example section section if included needs to be properly fleshed out where the reader can work out the example by hand
3. please add reference to Residual Quantization
with Implicit Neural Codebooks, ICML 2024 and other recent papers on VQ

**Questions:**

na

---

### Note · Authors · 2024-11-15

I have read and agree with the venue's withdrawal policy on behalf of myself and my co-authors.